

# Ice crystal images from optical array probes. Compatibility of morphology specific size distributions, retrieved with specific and global Convolutional Neural Networks for HVPS, PIP, CIP, and 2DS.

Louis Jaffeux[1], Jan Breiner[1], Pierre Coutris[1], and Alfons Schwarzenböck[1]

[1]Laboratoire de Météorologie Physique (UMR6016) / UCA / CNRS, Aubière, France

**Correspondence:** Louis Jaffeux (Louis.Jaffeux@uca.fr)

**Abstract.** The convolutional network methodology is applied to train classification tools for hydrometeor images from optical array probes. Two models were developed in a previous article for the PIP and 2DS and are further tested. Three additional models are presented : for the CIP, HVPS, and a global model trained on a data set that includes all available data from all four instruments. A methodology to retrieve morphology-specific size distributions from the OAP data is provided. Size distributions for each morphological class, obtained with the specific or global classification models, are compared for the ICE GENESIS data set, where all four probes were used simultaneously. The reliability and coherence of these newly obtained machine learning classification tools are demonstrated clearly. The analysis shows significant advantages of using the global model over the specific ones, in terms of compatibility of the size distributions. The obtained morphology-specific size distributions effectively reduce OAP data to a level of detail pertinent to systematically identify microphysical processes. This study emphasizes the potential to improve insights in ice and mixed-phase microphysics based on hydrometeor morphological classification from machine learning algorithms.

## 1 Introduction

The topic of ice crystal shapes has stimulated the imagination of cloud enthusiasts for centuries. One way to look at the morphology of solid hydrometeors is closely related to their growth history. These shapes reflect the fact that certain microphysical processes inside ice clouds do occur and, in some cases, even pinpoint their location in time and space (Pasquier et al., 2023). Particle morphology, therefore, helps infer the pathways available to form atmospheric ice. In particular, the shape of solid hydrometeors may be used to deduce environmental conditions that influence their growth, such as temperature, humidity, and turbulence levels within clouds.

Simultaneously, 3D particle geometry accounts for numerous properties of ice particles, such as fall velocity (Vázquez-Martín et al., 2021; Locatelli and Hobbs, 1974) , capacitance (Westbrook et al., 2008), scattering properties (Wyser, 1999), or melting behavior (Matsuo and Sasyo, 1981; Knight, 1979). In addition, further phenomena related to the natural differentiation of hydrometeors have to be mentioned. Extreme precipitation rates, electrification, or extended cloud lifetimes are some of



the visible consequences of ice particle interactions. These cloud-scale features ultimately play important roles in the global climate system and feedback on atmospheric conditions and thus cloud formation itself. An illustration of the intricate nature of these feedback mechanisms and their role in the emergence of new cloud-scale properties is given below to clarify the importance of morphology.

Dendritic crystals have high capacitance because of their 3D shape. They grow in the temperature region (-15 °C) where supersaturation reaches its potential maximum (Pruppacher and Klett, 2010). And, at last, their fall velocity is relatively low (Fukuta and Takahashi, 1999), resulting in prolonged residence time in their original growth region. Therefore, they are draining excessive amounts of water vapor. In contrast, graupel particles fall fast and exhibit rimmed surfaces, which inhibit depositional growth (Jensen and Harrington, 2015). In addition, they are particularly resistant to melting. As a result, rimed particles more effectively sediment towards the ground, ridding the cloud of some of its condensed water. When graupels and dendrites collide with one another, charge transfer happens (Emersic and Saunders, 2010). If enough of these charging events occur, cloud electrification reaches the point where atmospheric lightning is triggered. Lastly, lightning strikes play important roles in the climate system, such as the ignition of wildfires that release massive amounts of gas species (including greenhouse gases) into the atmosphere (Knorr et al., 2017). This example shows that different hydrometeor types contribute more effectively to different cloud functions, such as dissipation of supersaturation for dendrites and precipitation for graupel particles. In the presented example, these functions partly control the cloud life-cycle and fallout rate, two essential characteristics of clouds. In addition, the existence of differentiated particle types allows for the explanation of further phenomena, such as lightning.

Because ice morphology is at the core of ice cloud mechanisms, reporting and classifying ice particles holds an important place in the development of cloud microphysics. For airborne in-situ measurements, morphology is often limited to qualitative information because of instrumental limitations (Zhu et al., 2015; McFarquhar et al., 2007). However, quantitative size distributions are produced by analyzing the data obtained with optical array probes (OAP). Due to their poor resolution, extracting morphological information from OAP images was traditionally too difficult for feature-based approaches and very limited when performed manually because of the large number of images produced by these probes. However, a trained eye is able to distinguish particle shapes and assign the proper morphology to the majority of the produced 2D images. Feature-based approaches have tried to capture this human ability by using relevant geometric measurements on images (Duroure, 1982; Rahman et al., 1981) with limited success and extensive treatment speed (Praz et al., 2017). With the advent of artificial intelligence, this human ability can finally be emulated (Krizhevsky et al., 2012) with a neural network architecture called convolutional neural networks (CNN). Jaffeux et al. (2022) and Zhang et al. (2023) achieved reliable automatic classification for OAP images using this method. The current article builds upon the methodology and CNN models developed in the earlier study by the same authors.

Beyond the initial development of classification methodologies, comparisons of coherent observations, obtained with different instruments, are lacking. This gap in research represents a significant opportunity for data set exploration and validation of classification tools. While multiple OAPs are often mounted on research aircraft in order to cover complementary size ranges with some overlap, this study proposes a methodology to separate particle size distributions into morphology specific size distributions for each probe. Such size distributions, produced with OAP data, were previously reported in Jaffeux et al. (2023a),





but this is the first time they are presented in a strictly scientific article. Furthermore, a new single classification tool is trained
and used to process images from four different instruments. These spectra cover most of the upper range of the particle size
spectrum (from 300 $\mu$m up to 1.9 cm) and overlap largely in the special case of the ICE GENESIS data set, which contains
OAP data sampled simultaneously with the four mostly used OAP instruments (2DS, HVPS, CIP, and PIP). This instrumental
setup, combined with the machine learning classification tools, provides a first basis to analyze and compare particle size dis-
tributions of differentiated ice morphologies across most of their size range. The use of general CNN models versus limiting
the training set to only images of each specific probe is a key focus of the article.

The goal of this study is to demonstrate the capabilities of recently acquired set of classification tools. To achieve this
objective, 5 trained CNN models are presented and applied on the ICE GENESIS data set. The resulting morphology specific
particle size distributions are then compared to assess their compatibility.

The procedure that leads to quantitative estimates of morphology specific particle size distributions for different OAP instru-
ments is detailed in the first section. Therefore, two already existing classification tools, one for the 2DS and one for the PIP,
are presented, tested, and improved for the ICE GENESIS data set. Then, two additional CNNs that were developed for the CIP
and HVPS are quickly presented. Subsequently, the extracted morphological information and the quantitative bin concentration
estimation are combined.

In a second part, the data set is briefly presented in terms of the encountered thermodynamical conditions and total OAP data.
After that, the compatibility and coherence of the five classification tools are explored on this data set by studying morphology
specific size distributions for each class.

The final section provides a summary of the content of the article, conclusions on the size distribution analysis, including
potential improvements and recommendations, and finally, a discussion on the benefits of the use of the developed tools for the
community.

## 2   Methodology

CNN were able to solve the problem of shape recognition by reaching human levels for single images (Krizhevsky et al.,
2012). Nonetheless, these algorithms are not flawless, especially because they rely on manually gathered data sets. Faulty
class attributions may stem from the arbitrary definition of image classes or the possibility of encountering undefined particle
types in acquired data. Understanding that CNNs are black-box algorithms whose mistakes are difficult to decipher is the
main motivation for their testing. It is indeed necessary to address this weakness and evaluate the quality of the extracted
morphological data. Jaffeux et al. (2022) developed two classification algorithms for the precipitation imaging probe (PIP) and
the 2D-stereo (2DS) probes. To these 2 instruments the CIP and HVPS have been added.

The CNN structure used in this study is similar to AlexNet (Krizhevsky et al., 2012). It consists of two parts : a feature
extractor, and a classifier. The first part takes a fixed size image as input, and follows a hierarchical structure of successive
convolution and subsampling layers. This part converts the initial image and the then created feature maps into smaller, sum-
marized, higher level feature maps. The convolution layer applies the dot product to the values of each pixel and its surrounding





in a 3 by 3 square and 3 by 3 filters (also called kernels), that are trained to account for abstracted features. Then, the subsampling layer reduces the obtained feature map to its more crucial information using 2 by 2 max pooling filter, thus dividing the number of pixels of the feature map by 4. Because it does not provoke any increase in computational cost, the number of filters applied within each convolution layer is accordingly doubled as the feature map reaches deeper levels of the feature extractor.

The 2 two layer types are applied one after the other until the size of the feature maps reaches one. The number of convolution layers depends therefore on the size of the initial image. In the case of the ones obtained with the 2DS and HVPS, which both have 124 photodiodes, a size of 200 by 200 was used, resulting in 6 convolution layers. In the case of the CIP and PIP probes, which both have 64 photodiodes, a size of 110 by 110 was used, resulting in 5 convolution layers.

Finally, the classifier, a fully connected perceptron with a single hidden layer, is used to attribute classes to the combination

of the most abstracted features that were extracted. During the training, 20 % of the labelled data is randomly taken out for the final testing, 16 % for validation during training, and 64 % is used for training the filters weights and the synapses from the fully connected layers. The images are padded to the adequate input size. Then, they undergo a random flip operation (vertically, horizontally, both, or none) in order to produces more variety in the orientation of the particles without requiring pixel interpolation. Bayesian parameter optimization is performed for hyperparameter tuning, including the number of neurons

used in the classifier dropout (Srivastava et al., 2014) for each convolution and fully connected layers.

The goal of this first section is to describe the training data and the morphological classes associated with each CNN model, and to report the results of the training using confusion matrices and training reports obtained on test data sets.

## 2.1 Morphological classes

When considering clouds and their potential to produce precipitations, ice particles can be separated into 3 general types :

pristine crystals, intermediary particles and ultimate precipitating particles. Their respective number and mass size distributions and concentrations reflect the strength of the processes governing their appearance, growth and consumption. Because revealing and measuring these effects is the end goal of the developed tools, the classes defined in this subsection fall into these 3 categories.

1. Pristine crystals are single crystals formed through the initiation of ice particles that grow by the deposition of vapor.

Pristine crystals disappear through self-aggregation, scavenging by other particle types, and shape alteration via riming or secondary deposition regimes (in the special case of capped columns). These particles include plates and dendrites that are designed under a common class named hexagonal planar crystals (HPC). Because plates and dendrites grow in adjacent environments (Fukuta and Takahashi, 1999), they are considered a continuum here. Columns and needles (Co) are the other types of pristine crystals defined for the models.

2. Intermediary particles are formed from pristine crystals and the intermediary particles themselves. They grow through the aggregation of pristine crystals and intermediary particles, including self aggregation and secondary deposition. They are consumed by collection and riming. Based on the available OAP datasets, combinations of bullets and columns (CBC), complex assemblages (CA), fragile aggregates (FA), and capped columns (CC) are the defined classes corresponding to





this intermediary type. Combinations of bullets and combinations of columns are hardly differentiable with the coarse resolution and binary nature of OAP images. Both particle types follow the definition of intermediary types; for this reason, they were put together. Particles with complex shapes, sharp edges, exhibiting transparency, and often composed of spatial plates are commonly obtained, but only with the 2DS. The corresponding particles likely formed through the aggregation of pristine crystals and/or significantly grew by deposition in different environments. These observations motivated the definition of the CA class for the 2DS. In some cases, the individual elements composing an aggregate cannot be identified, either because these elements are too small with respect to the pixel resolution or because the elements are individually amorphous (e.g., aggregates of ice fragments). The FA class corresponds to these aggregate types in cases where the bounds between the monomers are relatively thin.

3. Ultimate precipitating particles are formed and grow by the riming and aggregation of any type of crystal, including self aggregation. The mass fall rate (or downward mass flux) of ultimate particles is thereby roughly constant. Two classes are defined to correspond to this definition : compact particles (CP) and rimed aggregates (RA). Both of these morphological classes are close in terms of shapes and mostly designate two archetypes of dense ice particles, CP being the most compact and RA being defined as particles that visibly contain several monomers.

All the classes defined above are summarized in Figure 1 with actual examples from the training data for each of the 4 probes. It can be noted that for the 2DS and CIP, a water droplet (WD) class was added. In addition to these classes, two artefact classes were defined for each of the SPEC instruments: a diffracted class for the 2DS (Dif) and a fragmented particle class for the HVPS (FP). These classes answer the fact that despite the use of inter-arrival time algorithm (Field et al., 2006), and treatment of diffracted images (Vaillant de Guélis et al., 2019), some of these artefact images are still found within OAP images. All training data sets and codes for the training can be found in the GitHub repository (see electronic appendix). The 2DS, CIP, PIP, and HVPS training data sets are composed of 6561, 5163, 3281, and 4290 images, respectively. The building of these training data sets is the result of an iterative process of training, testing, and gathering more pertinent data. Appreciable number of images in each class is a not a necessary condition to allow the CNN model to train and, more importantly, generalize successfully.

As mentioned before, the entirety of the training data for each probe was gathered into a single data set with the aim to train a model adapted for the data of any OAP. Two noticeable adjustments were made in the making of this new data set : additional 2DS dendrite and hollow column images were added, capped columns for the PIP were added, and the RA images were included in the CP class. The CA, Dif, and FP morphological classes only include data from one probe (2DS for the CA and Dif and HVPS for SP). The consequences of the decision to keep these classes in a general model are discussed in the analysis section. In total, 21390 images constitute this "global" data set. This number can be put in perspective with the 24 720, 9000, and 33 300 images used in Przybylo et al. (2022), Schmitt et al. (2024), and Zhang et al. (2023), respectively, and which used transfer learning for similar number of classes with the same objective of classifying ice particle shapes.

Hereafter, the quality of the training of each CNN, associated with each data set, is demonstrated. The CIP, HVPS, and "global" data sets are essentially built with data from the ICE GENESIS campaign. For the three corresponding CNNs, the





| Classe Names | 2DS examples (width = 2 mm) | | | CIP examples (width = 2.75 mm) | | | PIP examples (width = 11 mm) | | | HVPS examples (width = 19.2 mm) | | |
|---|---|---|---|---|---|---|---|---|---|---|---|---|
| Compact Particles | | | | | | | | | | | | |
| Fragile Aggregates | | | | | | | | | | | | |
| Columns and Needles | | | | | | | | | | | | |
| Hexagonal Planar Crystals | | | | | | | | | | | | |
| Rimed Aggregates | | | | | | | | | | | | |
| Combinations of Bullets and Columns | | | | | | | | | | | | |
| Complex Assemblages | | | | | | | | | | | | |
| Capped Columns | | | | | | | | | | | | |
| Water Droplets | | | | | | | | | | | | |

**Figure 1.** The nine morphological classes defined for the classification algorithms for the 2DS, CIP and PIP (Jaffeux et al., 2022).

confusion matrices and training reports are shown and shortly described. For the 2DS and PIP algorithms, that were obtained in Jaffeux et al. (2022) with data acquired prior to the ICE GENESIS campaign, additional specific testing and assimilation are
160 presented instead.

## 2.2 Additional CNNs : Cloud Imaging Probe (CIP), High Volume Particle Spectrometer (HVPS), global model

### 2.2.1 CIP

The CIP has a 25 $\mu m$ resolution and 64 pixels, providing measurements of hydrometeors in the 12.5 to 1600 $\mu m$ range. As for any OAP instrument, the upper size limit can be extended at the cost of measurement statistics and bias in undersizing truncated
particles, when reconstructing those. The 2DS classes are used except for the CA class, which was characterized by some level of transparency which could not be found within available CIP images. The same training methodology was used for the CIP CNN as for the PIP and 2DS ones, see Jaffeux et al. (2022). The CIP CNN was trained exclusively on ICE GENESIS data, as



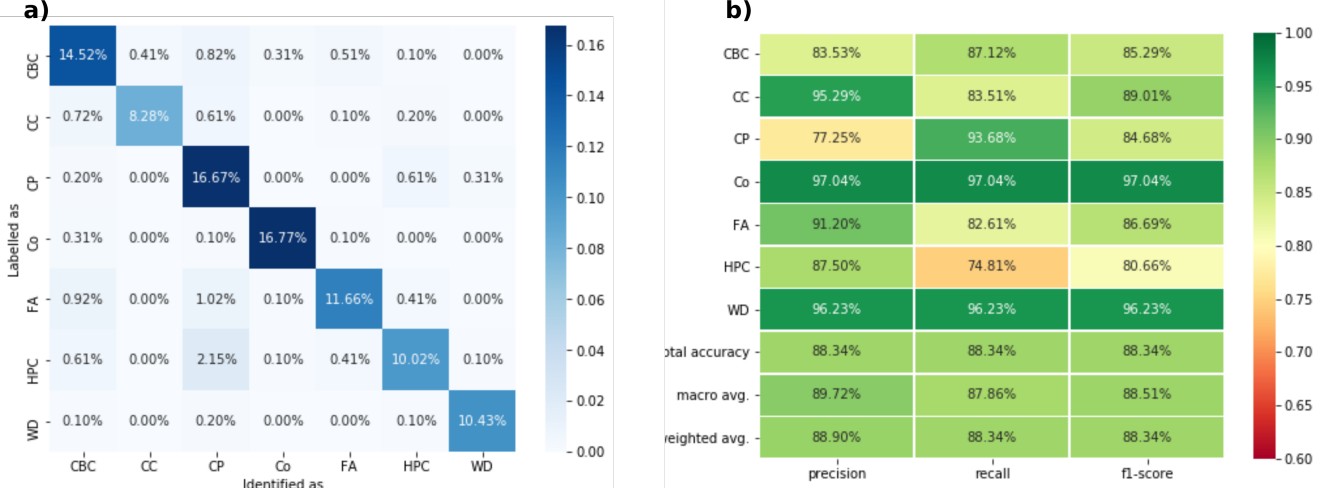

**Figure 2.** a) Confusion matrix for the CIP CNN. b) Training report for the CIP CNN.

opposed to the PIP and 2DS CNNs. The associated training confusion matrix and training report can be found in Figure 2. The most noticeable confusion happens between CP and HPC, but this confusion happens at the expense of the precision of the CP class, which is a good outcome considering the reasoning held in the previous subsection. The total accuracy is slightly below 90 %. It can be noted that both Co and WD classes are very well recognized.

### 2.2.2 HVPS

The HVPS has 150 $\mu m$ resolution and 128 pixels, providing measurements of hydrometeors in the 75 to 19200 $\mu m$ range. For the HVPS, all defined PIP classes are used, with the addition of the FP class. The HVPS CNN was trained exclusively on ICE GENESIS data, similarly to the CIP CNN. The resulting confusion matrix and training reports are shown in Figure 3. Few HPC were found within the HVPS data sets (318), leading to low precision for this class. The most noticeable confusion happens between aggregate classes, in particular the CNN identifies some RA and CBC as FA. The total accuracy is around 85 %.

### 2.2.3 Global model

On the validation set, accuracy reached 96.8 % and testing resulted in the following confusion matrix and training reports (Figure 4). Relatively low confusion was found between the FA, CBC, and CP classes on the one hand, and between CP and HPC on the other hand. The quality of the obtained model matches that of the specific models. Consistently with the specific models of 2DS and CIP, Co and WD are well recognized in the test data set.





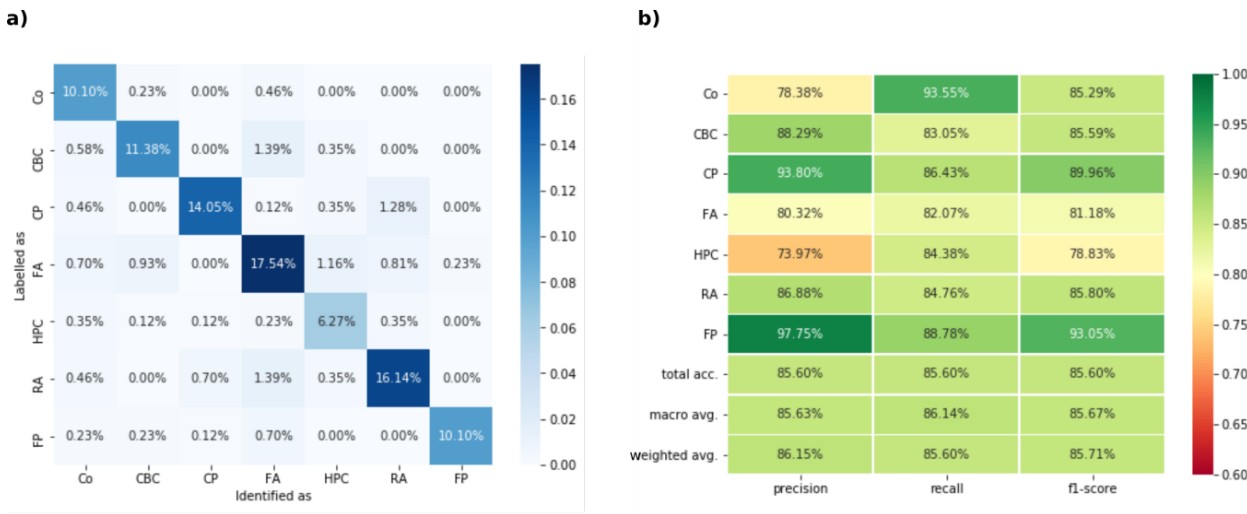

**Figure 3.** a) Confusion matrix for the HVPS CNN. b) Training report for the HVPS CNN.

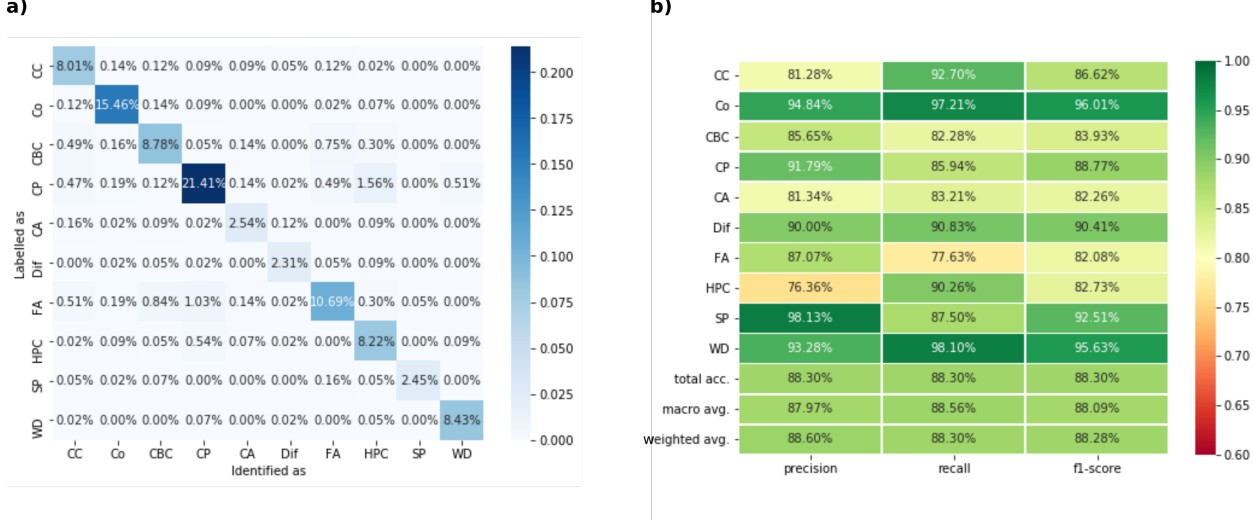

**Figure 4.** a) Confusion matrix for the All OAP CNN. b) Training report for the All OAP CNN.





### 2.3 CNN specific test and data assimilation

In order to specifically evaluate the performance of the previously trained algorithms (for PIP and 2DS) and improve their representation for each class, 200 images were manually extracted from the whole ICE GENESIS data set for each class for the PIP and 2DS, in order to test both CNNs with respect to their accuracy concerning each morphological class. In case the algorithm would show important amounts of error, data assimilation has to be performed to hopefully resolve the issues. The authors strongly encourage the manual inspection by the reader of the 200 images extracted for each class as their selection is inherently subjective. Such manual verification is required to help understanding of the predictions of the CNN models in subsequent analysis.

#### 2.3.1 2DS

Figure 5a describes the results of the specific testing. While columns are perfectly recognized, and generally most classes are well identified, some porosity between CBC and FA on the one hand, and between HPC and CP on the other hand, are found in reasonably limited amounts. However, the model iteration completely fails to recognize WD. Comparison between the test data and training data shows that the test droplets are vertically elongated, while the training ones were either horizontally elongated or spherical. This difference stems from the processing methodology used at the LaMP, which is the set up of a high default true air speed (TAS) for OAPs rather than online direct update from the plane's central computer. While size parameters are corrected, since the default TAS is higher than the real one, images are not resized accordingly. In other words depending on the plane's speed, different levels of 2D image deformation were experienced for the droplets. Since water droplets have very characteristic shapes, the CNN is highly disturbed by this change and does not recognize water droplets from the ICE GENESIS data at all. Finding a remedy to this problem is required, not only because identifying WD is essential to the microphysical analysis, but also because their presence currently harms the precision of the CP and HPC classes.

The tested data were assimilated into the training set in order to obtain a model that is performing even better with the ICE GENESIS data. A substantial improvement for water droplet recognition was required and thus a supplementary amount of 700 images of water droplets of all sizes were added to the training data set. The results of this second inspection are presented in Figure 5b. The final testing results improved significantly : the overall accuracy rose from 60 to 82 %, with 100 % recall for WD and 92 % accuracy. The results for the HPC class changed considerably as well. While before the assimilation many additional images would have been misidentified as HPC, after the assimilation, even if the class-specific recall is lower (from 73 to 52 %) the accuracy is higher (from 42 to 88 %). This shows that the updated version of the model circumvents one of the difficulties humans are also subject to, namely : a heavily rimed plate and a spherical graupel are difficult to distinguish from OAP images, with the lack of surface information being one of the reasons. While it is acceptable to classify heavily rimed HPC as CP, the inverse classification is problematic for the study of microphysical processes. Overall, the above test improved the algorithm's predictions and its scientific interpretability.



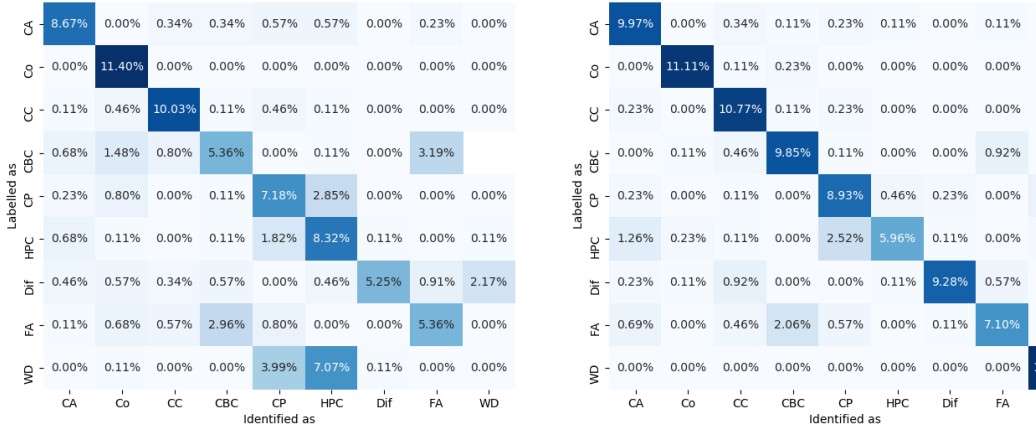

**Figure 5.** Results of specific evaluation for the 2DS classes in number (100 images per class). The horizontal axis describes user classification and the vertical one the CNN algorithm results. a) First evaluation. b) after assimilation of the data from a) and additional water droplets images.

### 2.3.2 PIP

Figure 6a describes the results of the class specific testing. Co and CBC have acceptable recall and precision. The expected porosity between RA, FA, and CP is confirmed. The porosity occurs at the expense of the FA class, which is an acceptable outcome. A result that has to be mentionned is the low accuracy of the HPC class (59 %) and the inclusion of a high number of CP in the HPC predictions. In order to try and rectify this confusion, initial testing data were assimilated and produced the results shown in Figure 6b. The overall accuracy improved from 66 to 72 %. Porosity between FA and CBC and between

CP and RA is still considerable. A significant number of images of columns of only a few pixels are sometimes identified as FA. Nevertheless these errors are acceptable because they do not harm the ability to make physical interpretations since they respectively emanate from the same microphysical processes. Finally, the most problematic class in the original testing, HPC, was fixed after data assimilation with an accuracy of 91 % and a recall of 72 %.

### 2.4 Computation of number and mass size distributions for different morphological classes

The four probes have pixel resolutions of 150, 100, 25, and 10 $\mu m$, for the HVPS, PIP, CIP, and 2DS, respectively. The 2DS and HVPS have arrays of 128 photodiodes and the CIP and PIP of 64. Habit recognition requires the definition of minimum particle size this lower size threshold was set to 20 pixels for the HVPS, PIP, and CIP and 30 pixels for the 2DS. This difference is motivated buy the high number of artifact particles that was obtained from 2DS images with the 20 pixel requirement. In addition, truncated images were excluded from the classification. The "entire-in" criterion provides the best shape recognition

capabilities, ensures the quality of the extracted 2D information, and it the original sampling volume computation formula,



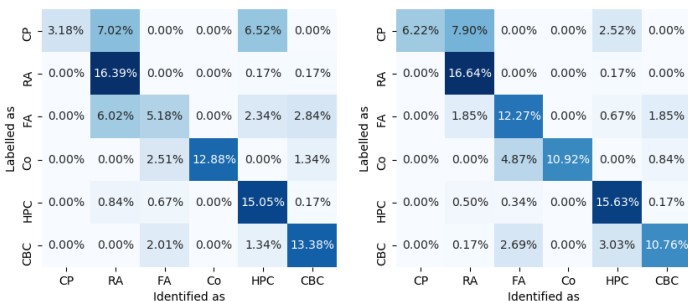

**Figure 6.** Results of specific evaluation for the PIP classes in number (100 images per class). The horizontal axis describes user classification and the vertical one the CNN algorithm results. a) First evaluation. b) After data assimilation.

providing an accurate calculation of particle concentrations (Knollenberg, 1970). Nonetheless, this criterion reduces the number of analysed images especially for size ranges nearing the length of the photodiode array, beyond which, only elongated particles oriented perpendicularly to the detection array are kept. Despite this artificial filtering, no upper size limit has been set for any of the probes used. In particular because this filtering heavily depends on particle shapes, the classification opens the possibility to even quantitatively explore this effect.

Since the classification tool is not yet integrated within the processing tools of OAP images at the LaMP (Leroy et al., 2016), single images are extracted, tagged with timestamps and identification keys, so that they can be merged with routinely processed image features such as perimeter, surface area, etc... After padding the raw images, they are processed by the classification tool, and each image is registered as one line in a table (pandas DataFrame) with its identification keys, timestamp, geometric features, and classification score for each class (the sum of which is normalized). The classification score is rounded so that it becomes a categorical binary array (one 1 and zeros). In addition, the aircraft data, with in particular altitude, temperature, humidity measurements, and position, is resampled and concatenated in additional columns. Estimation of the sample volume is necessary in order to link classification results to quantitative concentration values. Computing the sample volume, in the existing code, is a dynamic operation that takes into account all recorded particles within the time frame of 1 second and the operational time of the probes. No direct method can therefore retrieve a particle by particle sample volume. In the future, the classification tool has to be integrated within the existing feature extraction IDL routines. For now, a simple approach was developed to retrieve class specific number and mass size distributions which easily translate to number and mass concentrations. The method is detailed below and illustrated in Figure 7.

1. Particles are filtered : in time, space (including altitude), or by relevant physical parameters such as temperature. The result of this step is a collection of particles gathered under user-defined filters.

2. A binned $D_{max}$ is calculated for each particle. Particles are then grouped (summed) within bins corresponding to the bins used in the LaMP routine for particle size distribution (PSD) computation : bin width 10 $\mu m$ (100 $\mu m$) and first bin center is 5 $\mu m$ (50 $\mu m$) for the 2DS (PIP). The result is a pseudo class-specific PSD, uncorrected by the sample volume.





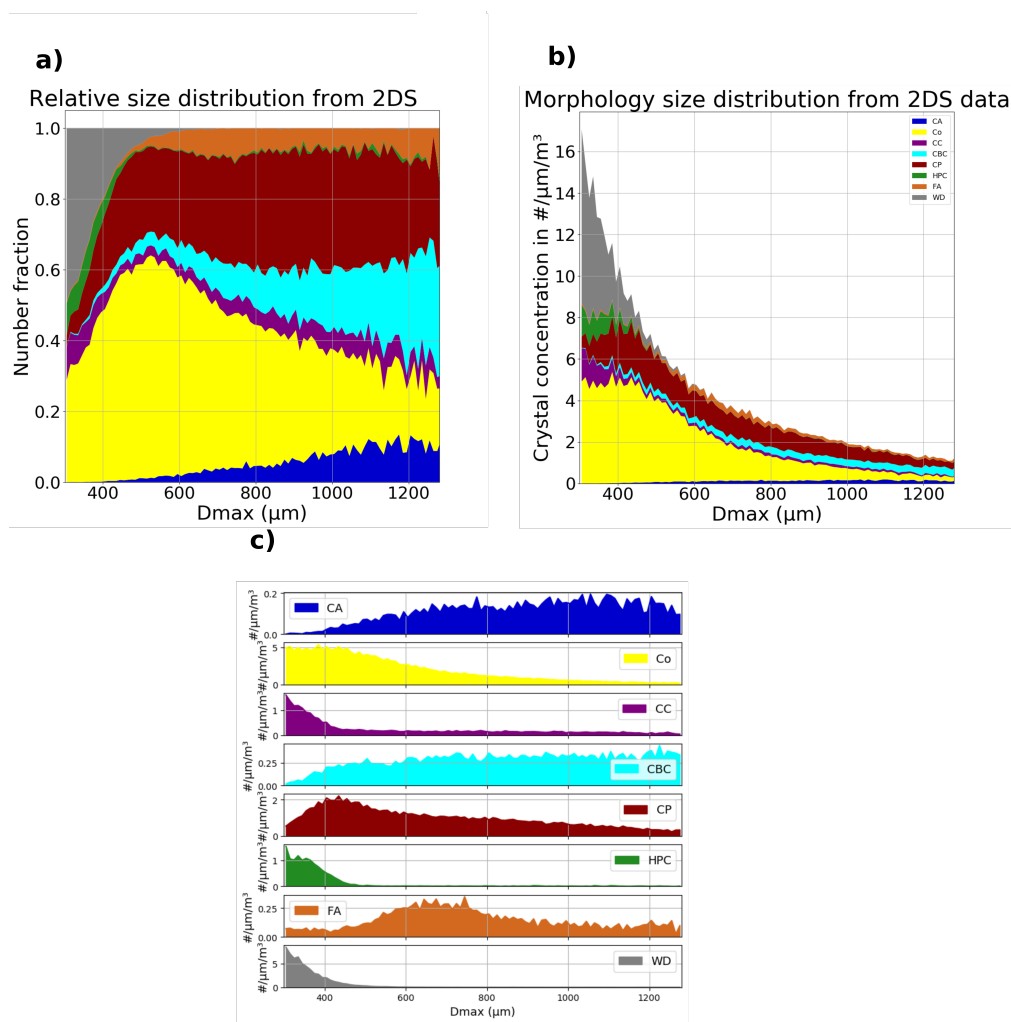

**Figure 7.** Plots of morphology specific size distributions from 2DS data obtained in a) Step 3, b) Step 4, and c) Step 5 of the computation of habit specific Particle Size Distribution (PSD). The data plotted corresponds to the data gathered during flights 6, 7, and 8 of the ICE GENESIS field campaign between +2 and -9 °C (see Section 3.1).




3. The number of images per class is normalized by the total number of particles in each bin, yielding a fraction of each type of particle in each bin (see Figure 7a).

4. After applying the filters (used in step 1). to the PSD obtained through the LaMP routine, a PSD corresponding to the obtained class-specific normalized PSD is obtained. By multiplying bin by bin the normalized class-specific pseudo PSD with its corresponding PSD, the class-specific PSD is obtained (see Figure 7b). The stacked PSD's envelope is the former PSD, obtained with the LaMP routine.

5. Plotting each morphology separately yield class specific PSDs such as in Figure 7c. And consequently, the sum of each PSD gives concentration values.

As a side note, the corresponding mass PSD can also be retrieved by using the estimated mass of each particle, through Baker and Lawson (2006) combined single parameter parametrization or mass laws specific to each particle type. However, this step is not within the scope of this paper.

# 3 Application of the methodology to the ICE GENESIS campaign

## 3.1 The ICE GENESIS Data set

The January 2021 ICE GENESIS field campaign took place in the Swiss Jura mountains, over the Lachaux-de-Fond airport (Billault-Roux, 2023). The primary objective of this airborne experiment is to document snow conditions in the 0 to -8 °C temperature range, in terms of hydrometeor number concentrations, sizes, and shapes. Snowfall environments were found by sampling winter frontal clouds, whose precipitation were enhanced through the orographic effect of the mountainous terrain. OAPs were used to obtain the required measurements with high reliability. 4 of the most commonly used OAPs were mounted on the SAFIRE ATR-42, including a 2DS, a 25 $\mu$m resolution CIP, a PIP, and an HVPS. The variety of resolution and size ranges of the probes provided measurements of hydrometeors in the 10 $\mu$m to 1.9 cm size range, with two extensive overlap regions between the 2DS and CIP and between the PIP and HVPS. In addition, it allowed having adapted resolution for particles of various sizes, meaning that, given the CNN models presented in Section 2, morphological recognition of particles was possible from 300 $\mu$m up to 19.2 cm. Results of the campaign documenting snow with these OAP measurements are presented in two conference papers (Jaffeux et al., 2023a, b).

The airborne RASTA W-band radar reflectivity vertical profiles for flights 6, 7, and 8 of the ICE GENESIS campaign are presented in Figure 8. 5 cloud segments have been selected, corresponding to rather deep frontal clouds that were sampled continuously. For segment 3, a warm front was sampled and for the other segments cold fronts. Together, they represent 7 flight hours, where the cloud tops reached temperatures below -20 ° C with bases nearing the melting point of water, meaning dendrites, plate-like, and columnar crystal types could be observed simultaneously with large snowflakes and rimed particles. In the present section, the data gathered within these periods are examined as a whole.

Figure 9 shows the flight distance corresponding to 1 degree temperature intervals. About 25 % of the total 3473 km were performed around -2 °C. The dendritic and platelike growth regions (between -20 to -12 °C, and between -12 to -8 °C,



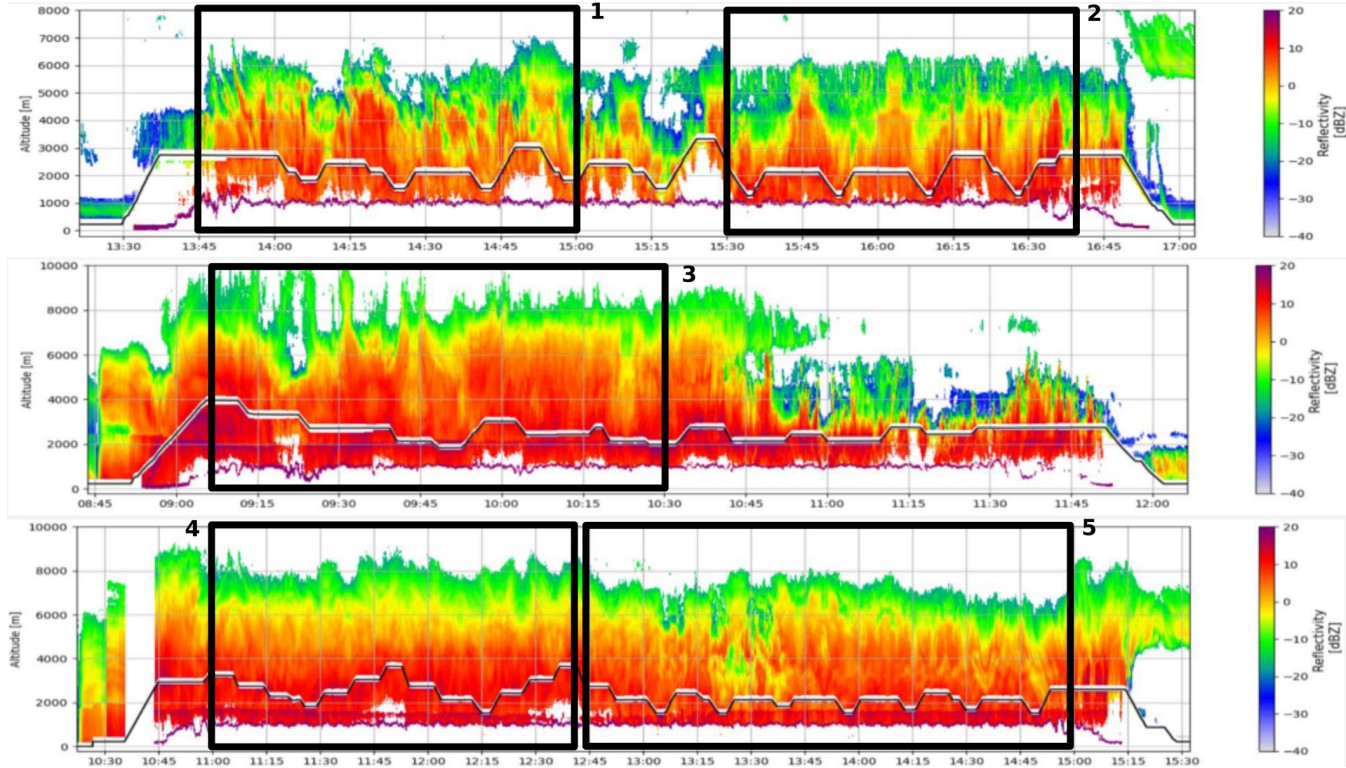

**Figure 8.** RASTA reflectivity time series for each flights and identification of the 5 selected cloud segments, symbolized by the black rectangles and numbered accordingly.

respectively) were not sampled, but some large precipitating crystals of these types were reported during the flights, and some smaller ones could be observed within larger aggregates. However, the columnar growth region (-8 to -5 °C) was well explored. Supercooled liquid droplet pockets were marginally detected by the Cloud Droplet Probe outside the melting layer's vicinity.

The total PSDs were obtained with the all-in method and are presented in Figure 10 for each OAP. With the exception
of the ends of the spectra of both DMT instruments (CIP and PIP), where values tend to fall off before $D_{max}$ reaches the photodiode array length, the four probes agree well over their overlapping range. In addition, the junction of both pairs of spectra is straightforward. In summary, all four spectra are well compatible.

For the 2DS, only the horizontal channel was processed. A total of 410 882, 578 164, 986 965, and 257 103 images met the size requirements and all-in criterion for the 2DS, CIP, PIP, and HVPS, respectively. They were analyzed by both specific and
global CNN models. Using the methodology described in subsection 2.4, they were combined with the total PSD presented in Figure 10. The obtained PSDs are described in the next subsection. For each morphology-specific PSD, total concentration values were obtained by summing all the bins of each spectrum. These concentrations were then normalized to give the pie charts shown in Table 1. They reflect the type of ice particles that were encountered within the data set for each probe and identified with the global and specific models presented in subsections 2.2 and 2.3.




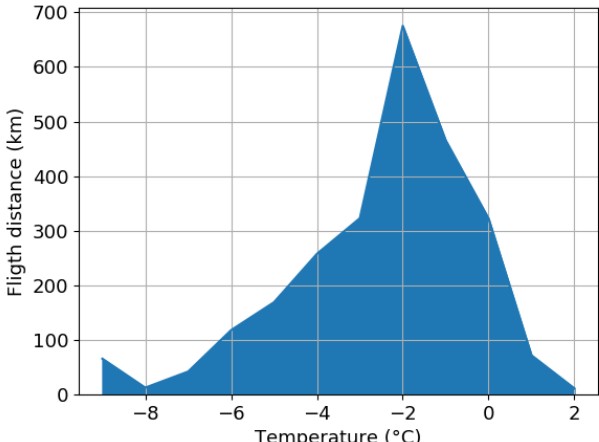

**Figure 9.** Flight distance during legs as a function of temperature during the 5 selected flight segments shown in Figure 8.

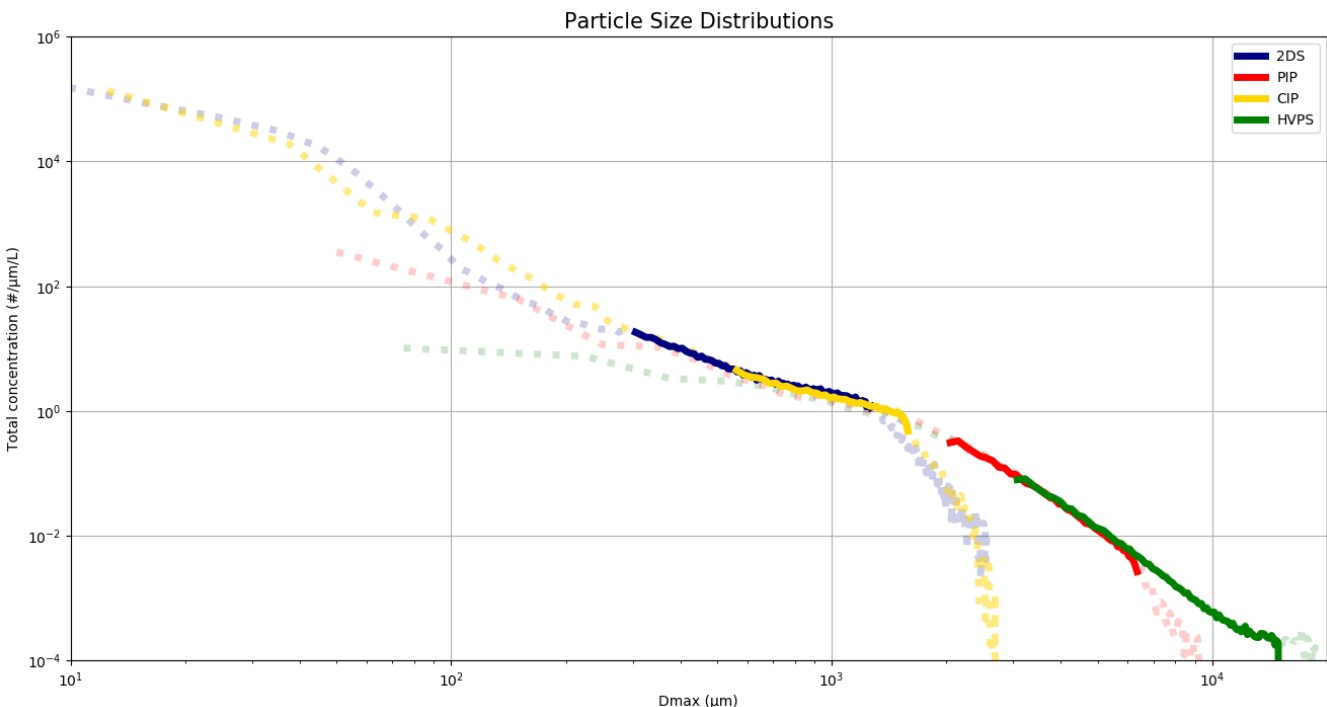

**Figure 10.** Total particle size distributions for each OAP obtained with the ICE GENESIS data set. Transparency and dots are used to distinguish the regions of each spectrum below the size threshold and above the photodiode array length.



For the fine resolution probes (2DS and CIP), Co is the most frequent hydrometeor type, followed by CP. For the coarser resolution probes (PIP and HVPS), CP, RA, and FA are the most frequent types. Both the specific and global models exhibit the same general trend: as pixel size gets larger, the relative concentration of Co decreases and the relative concentration of FA increases. The specific and global models are consistent for the 2DS, CIP, and HVPS, keeping in mind that the RA class is included in the CP one for the global model. Concerning the PIP, significantly fewer FA are identified by the global

model in comparison with the specific model. This discrepancy likely stems from different class definitions inferred from the corresponding training sets. This disparity shows that the definitions of the FA, CP, and RA classes differ between the PIP-specific and global models. The consistency of the classes across various instruments is, in this regard, a significant benefit of the global model.

## 3.2    Analysis of morphology specific size distributions

In the present subsection, particle size distributions obtained with specific and global models are compared for each class or set of classes. The objectives are to assess the compatibility of the distributions obtained for each instrument, and to evaluate which of the specific or global models yields the better results.

### 3.2.1    Compact Particles and Rimed Aggregates

Images of quasi-spherical, densely rimed graupel-like particles make up the CP class. RA images are rimed in a similar way,

but their shapes suggest underlying aggregates. This leads to a significant porosity between these two categories. These two classes are designed to reveal the occurrence of riming and/or aggregation in clouds. The CP and RA categories of the HVPS and PIP may contain more particles than their 2DS or CIP counterparts because unrimed aggregates cannot be distinguished from these particles at higher pixel resolutions. Therefore, the PIP and HVPS classes should likely overestimate the number of these particles compared to the 2DS and CIP ones. Figure 11 presents the CP and RA specific particle size distributions for all

4 probes and both models.

    The results obtained with the specific models for the RA class (Figure 11a) show some consistency between the PIP and HVPS. The agreement between 2DS and CIP for the CP class (Figure 11b) is satisfactory, but the agreement between PIP and HVPS is significantly worse. At 3 mm, the HVPS curve is one order of magnitude above the PIP, and this difference only grows larger with increasing particle size. The curves corresponding to the 2DS and CIP do not match well with the curves

from the PIP and HVPS. The former beginning at values one order of magnitude below the highest CIP bin size. This can be due to the absence of a RA class for the 2 lower resolution probes. The addition of the RA and CP class for the HVPS and PIP was performed and yielded (Figure 11c). This operation only marginally improves the coherence of the two sets of curves, contradicting the earlier hypothesis that the HVPS and PIP should overestimate the number of these particles. However, there is one element of explanation left to investigate: the class definition for each model. Every model constructs an abstraction

of what a CP or RA ought to be based on the training sets linked to the particular classes as well as in comparison to the other classes that have been defined. As can be seen in Figure 11d, the possibility to fill in the size range gap for the CP class





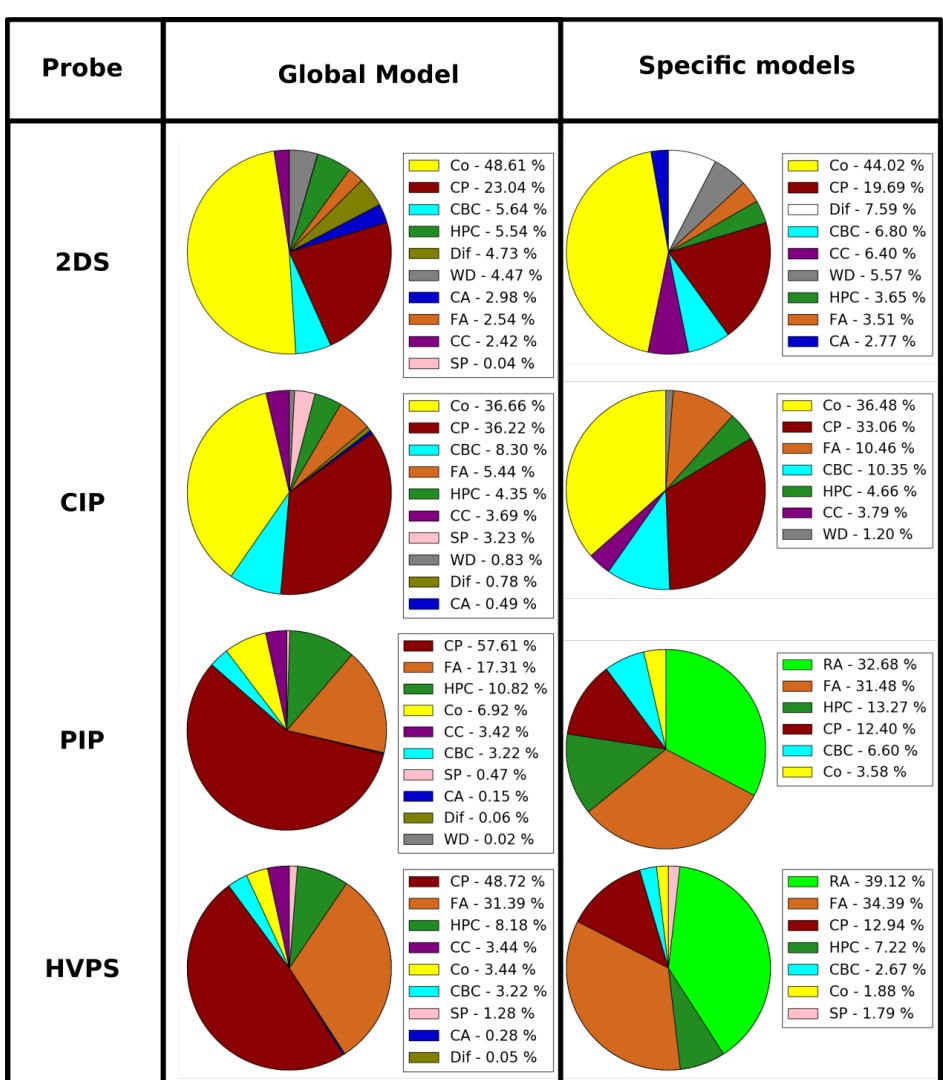

**Table 1.** Pie charts representing the hydrometeor fractions for total number concentration, by morphological class, and for each OAP identified with the global and specific models from the ICE GENESIS data set.

**Figure 11.** Total particle size distributions for each OAP obtained with the ICE GENESIS data set for the CP and RA classes, and identified with specific models (a, b, and c) and the global model (d). The black points show the distribution points above the photodiode array length.





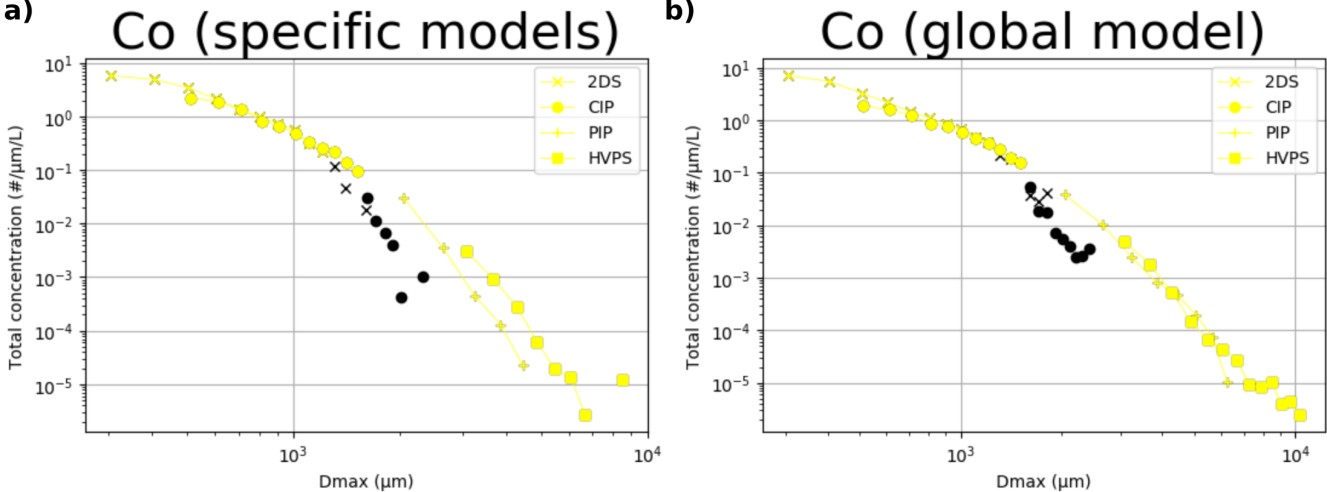

**Figure 12.** Total particle size distributions for each OAP obtained with the ICE GENESIS data set for the Co class, and identified with specific (a) and the global model (b). The black points show the distribution points above the photodiode array length.

(that now includes the former RA class) improves significantly when a single model is utilized to identify the images from the various probes. However, the overlaps between the two pairs of OAP is slightly worse in comparison with the specific models.

### 3.2.2 Columns

Co consists of images of single columns or needles. This class is meant to identify deposition growth in the columnar regime. Throughout the training and testing phases, the precision of this class was remarkable for every model. However, elongated particles are not necessarily columns, and with coarse pixel resolution, misidentification of images is possible if the training set does not include enough elongated particles in other classes, such as FA. The corresponding specific PSDs are presented in Figure 12.

Figure 12a presents the Co-specific particle size distributions obtained with the specific models. These results are consistent across the 2DS and CIP, but the PIP and HVPS disagree strongly with values one order of magnitude above the PIP ones. The curves from the CIP or 2DS match relatively well with the PIP or HVPS ones, and it is difficult to figure out whether the PIP or the HVPS matches best with the higher resolution probes. Concerning the global model (see Figure 12b), both the 2DS-CIP and the PIP-HVPS pairs agree well on their respective overlap ranges. In addition, joining the highest CIP and lowest PIP size 345 bins is straightforward.





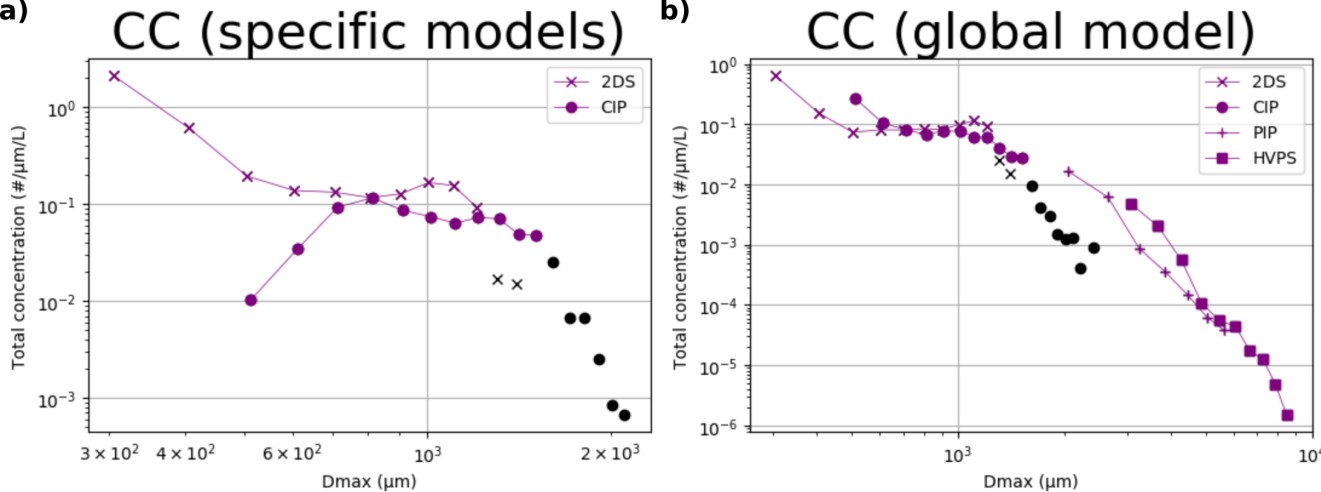

**Figure 13.** Total particle size distributions for each OAP obtained with the ICE GENESIS data set for the CC class, and identified with specific (a) and the global model (b). The black points show the distribution points above the photodiode array length.

### 3.2.3 Capped Columns

The CC class features images of capped columns that may appear as 'H'-shaped particles depending on their orientation. These particles are generally of sizes below 2 mm. Aggregates of a few columns can be mistaken for capped columns, especially with lower resolution probes. The PSDs corresponding to this class are shown in Figure 13.

For the specific models (Figure 13a), only the 2DS and CIP are available. Except for a few points, both distributions are very different, in particular for the smaller bins. Below 500 $\mu$m, the 2DS points are very high. The corresponding images show diffracted columns that look very similar to capped columns. Such diffraction patterns were already reported in Jaffeux et al. (2022). For the global model (Figure 13b), both CIP and 2DS distributions show similar shapes and agree above 600 $\mu$m. CC distributions are also now obtained for the PIP and HVPS. With respect to the 2DS spectrum, much fewer diffracted columns

are wrongly classified in the first bins. Important to note is the absence of HVPS images within the training set. Significantly more HVPS CC are found for size bins below 5 mm compared to the PIP. The visual inspection of these images revealed pictures of aggregates of a few columns or fragile aggregates with an H-shape, which can understandably be mistaken for capped columns, constituting legitimate identification errors. Similarly, for the PIP distribution, it is unlikely that 6 mm capped columns were encountered during the three ICE GENESIS flights. This implies that the rarely identified capped columns are

particles whose images look like capped columns rather than real capped columns. However, the total concentrations associated with this class are significant for the coarse resolution instruments (3.44 and 3.42 % of the mean concentrations for the HVPS and PIP, respectively—values from Table 1) above the CBC concentrations. For these two instruments, the CC class could be grouped with the CBC or FA classes. Alternatively, the global training set could be enriched through the assimilation of some





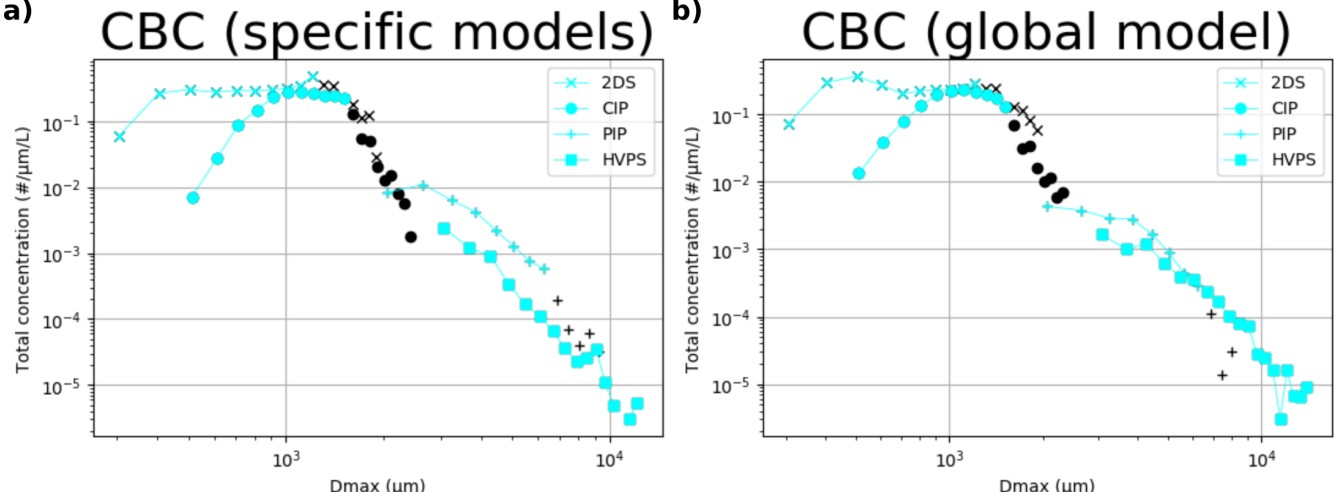

**Figure 14.** Total particle size distributions for each OAP obtained with the ICE GENESIS data set for the CBC class, and identified with specific (a) and the global model (b). The black points show the distribution points above the photodiode array length.

of the misclassified particles. This may result in increased precision and giving the classification algorithm the ability to better distinguish capped columns from other particle types for the PIP and HVPS.

### 3.2.4 Combinations of Bullets and Columns

Images of aggregates of columns and bullet rosettes can be found in the CBC class. Members of this class can only be classified by identifying individual columnar monomers in a larger image. Higher resolution images provide a more precise view of individual columnar monomers, aiding in accurate classification. This means that the pixel resolution has a major influence on this class. CBCs that are detected by high resolution probes are typically aggregates of smaller columns than those that are imaged and detected at coarser resolution. Figure 14 shows the corresponding PSDs.

The results of the specific models are presented in Figure 14a. For the 2DS and CIP, plateaus are reached above 400 and 1000 $\mu$m, respectively. Below 1 mm, the difference between the two curves is likely a consequence of the previously mentioned resolution effect, considering the 2DS and CIP resolution are significantly different (10 and 25$\mu$m, respectively). The PIP bin concentration values are higher for the PIP over its optimal range compared to the HVPS, which can be similarly explained. However, the PIP and HVPS resolutions are relatively close (100 and 150 $\mu$m, respectively). A possible explanation is the difference between the two definitions of the CBC class for each probe. The global model shows similar results, but it brings the PIP and HVPS much closer to one another in particular above 5 mm.

The results of the specific models are presented in Figure 14a. For the 2DS and CIP, plateaus are reached above 400 and 1000 $\mu$m, respectively. Given that the 2DS and CIP resolutions are significantly different (10 and 25$\mu$m, respectively), the difference between the two curves below 1 mm is likely due to the resolution effect discussed above. The PIP bin concentration values



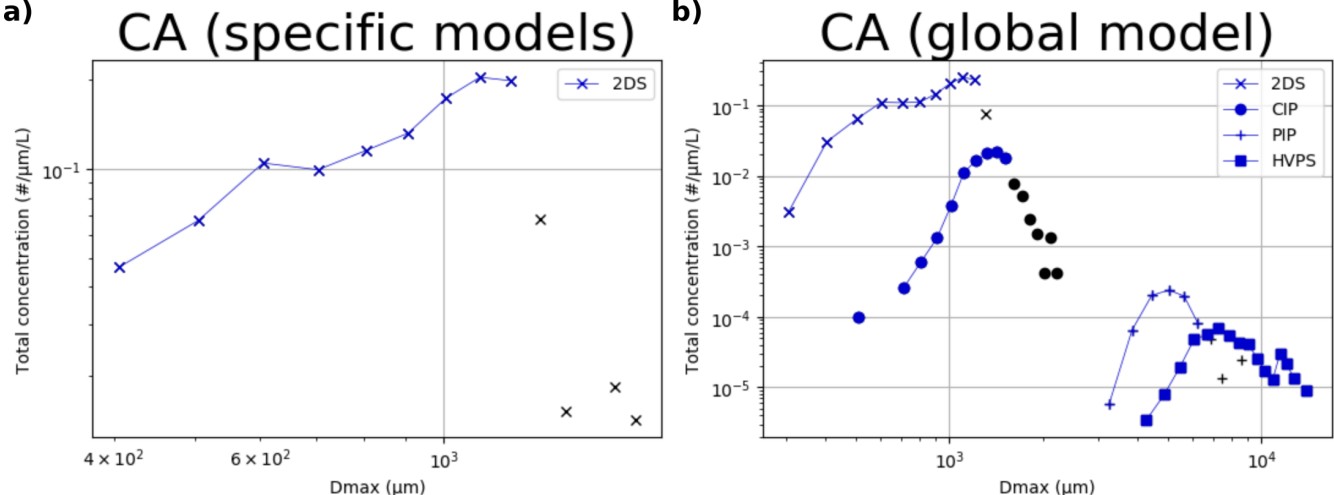

**Figure 15.** Total particle size distributions for each OAP obtained with the ICE GENESIS data set for the CA class, and identified with specific (a) and the global model (b). The black points show the distribution points above the photodiode array length.

are higher for the PIP over its optimal range compared to the HVPS, which can be similarly explained. However, the PIP and HVPS resolutions are relatively close (100 and 150 $\mu$m, respectively). A possible explanation is the difference between the two definitions of the CBC class for each probe. Comparable results are obtained with the global model for the 2DS, CIP, and

HVPS. However, the values of the PIP distribution decrease by a factor of 2, bringing it much closer to the HVPS distribution, in particular above 5 mm. This suggests that the differences between the PIP and HVPS using specific models may be due to variations in how the CBC class is defined for each probe.

### 3.2.5 Complex Assemblages

CA is a class of ice particles whose images show sharp edges and transparency and frequently display multiple plates or sector

plates. The corresponding particles indicate the occurrence of deposition in highly saturated environments and possibly the aggregation of plates and dendrites. The high level of detail that is required for their identification was only found in 2DS images. The PSDs obtained for this class are shown in Figure 15.

Out of the four specific models, only the one trained for the 2DS possesses this class (see Figure 15a). For the global model (Figure 15b), this class appears for each probe. Concerning the 2DS global model output, it corresponds to the specific model

almost point by point. For the CIP, a similar curve is produced but with much lower concentration values and at larger bin sizes. The inspection of the corresponding images is surprising as they share a common feature with the 2DS defined class : the sharpness of their edges. The CA class was therefore neither discarded nor assimilated into another class in the global model. For PIP and HVPS, the distributions are bell shaped curves centered at 5 and 7 mm, respectively. These curves exhibit a staggered pattern comparable to the CBC curves. This implies that pixel resolution has a similar effect on the CA class. For





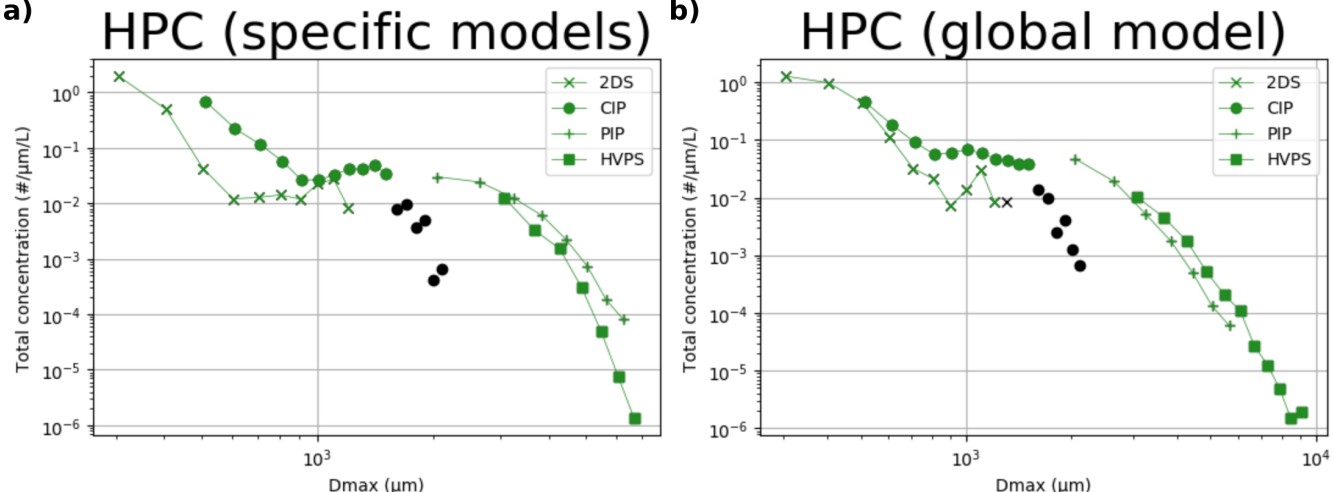

**Figure 16.** Total particle size distributions for each OAP obtained with the ICE GENESIS data set for the HPC class, and identified with specific (a) and the global model (b). The black points show the distribution points above the photodiode array length.

probes other than the 2DS, the CA class is, however, very marginally seen within the present data set for probes other than the 2DS, accounting for 2.98, 0.49, 0.28, and 0.15 % for the 2DS, CIP, PIP, and HVPS total number concentration, respectively. Significant uncertainty is associated with the identification of CA particles with probes other than the 2DS. However, the shapes of the curves obtained are encouraging with respect to the potential to identify particle morphology in poorly resolved images beyond human capabilities. In this instance, a model trained on high resolution images was used on lower resolution images,

yielding satisfactory results. These findings suggest that the identification of CA particles using probes other than the 2DS may be feasible with further research.

### 3.2.6   Hexagonal Planar Crystals

HPC is the class of single plates and dendrites. The sixfold symmetry and the planarity of the corresponding particles are the two defining features of this class. Learning these characteristics was found to be particularly challenging and showed relatively

low precision or recall during the testing for every model. As a reminder, recall was low for 2DS and CIP, precision was low for the HVPS, and both were low for the PIP. With respect to its dependency on pixel resolution, the HPC is a special case. A minimum size greater than the minimum threshold for each probe might be needed to accurately identify them because of the effect of pixelization on sharp edges of a few pixels in length. However, with an adequate pixel number, resolution should have no impact. The PSDs obtained for the HPC class are shown in Figure 16.

For the specific models (see Figure 16a), a large difference is noted between the 2DS and CIP below 1 mm, with the CIP being about one order of magnitude above the 2DS. Despite their divergence at large bin sizes, the PIP and HVPS curves are generally close, with the PIP values being higher than the HVPS. This discrepancy can be attributed to the high recall and low





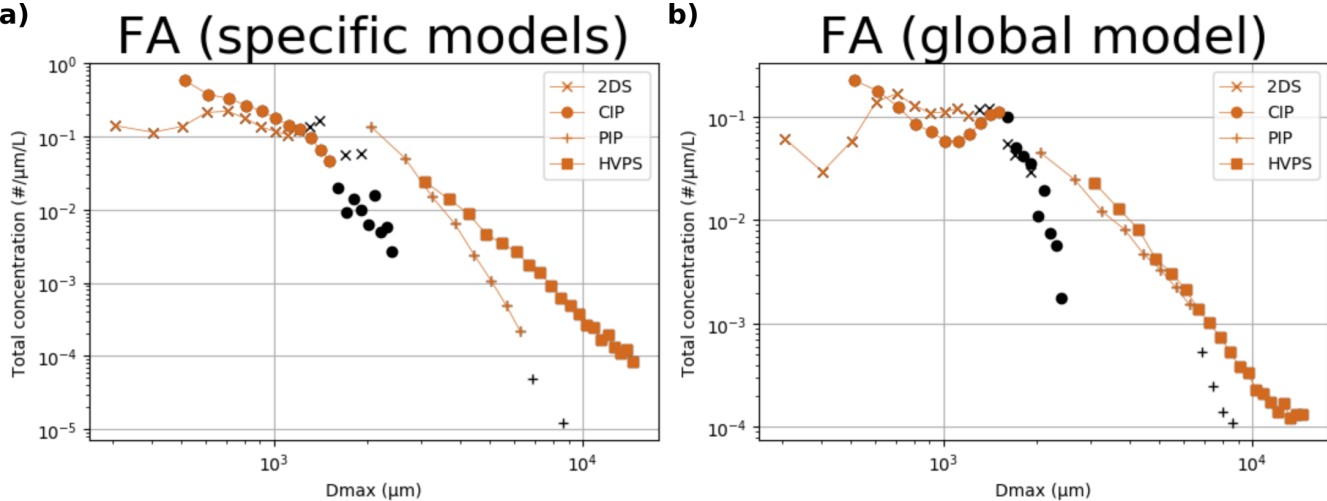

**Figure 17.** Total particle size distributions for each OAP obtained with the ICE GENESIS data set for the FA class, and identified with specific (a) and the global model (b). The black points show the distribution points above the photodiode array length.

precision on the HVPS testing set, indicating a possible overestimation on the HVPS side. For the global model (Figure 16b), the CIP and 2DS curves are much closer, with the few first CIP values decreasing and those of the 2DS increasing slightly in
comparison with the specific models. The HVPS and PIP curves are also relatively closer to one another. Overall, the results suggest that the global model may provide a more accurate identification for the HPC class compared to the specific models. However, the visual inspection of the classified images shows that improvements are still desired to reduce the number of particles that were misclassified in this class. Further data assimilation could potentially enhance the accuracy of the global model for classifying HPC particles.

### 3.2.7 Fragile Aggregates

Weakly linked aggregates with monomers that cannot be recognized as specific crystal types make up the FA class. Because it is likely to contain CBC or CA particles, whose details could not be rendered for low resolution probes, this class depends on pixel size. Figure 17 displays the PSDs that were obtained for the FA class.

For the specific models (Figure 17a), the 2DS and CIP agree for bin sizes above 500 $\mu$m to some extent. Similarly to the
results obtained for CP with the particular models, the PIP and HVPS curves show a significant divergence. The 2DS/CIP and HVPS/PIP distributions are difficult to join in agreement with the previously mentioned resolution dependency of this class. The classification with the global model (Figure 17b) considerably alters the shape of the CIP distribution, with a minimum at 1 mm. The 2DS size distribution is only significantly lower in the first bins compared to the specific curve. Finally, the PIP and HVPS curves are closer using the global model. This result is reassuring considering the similar pixel resolution between





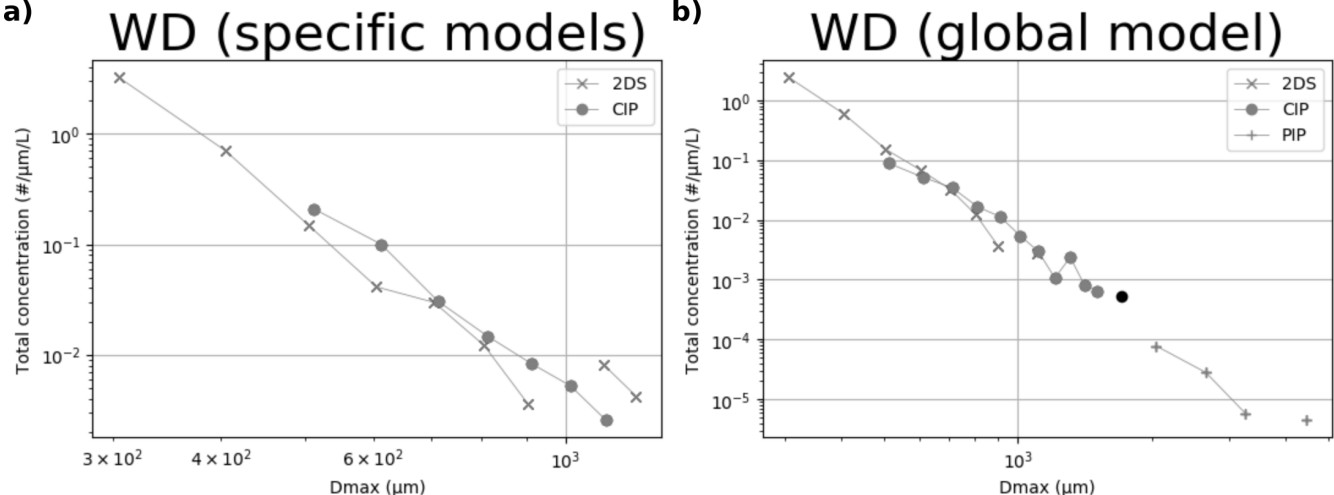

**Figure 18.** Total particle size distributions for each OAP obtained with the ICE GENESIS data set for the WD class, and identified with specific (a) and the global model (b). The black points show the distribution points above the photodiode array length.

both instruments. For the FA class, which is more loosely defined compared to the other class, much of the slight differences between the different morphology specific PSDs might have implications for the FA PSD.

### 3.2.8 Water Droplets

The WD class is composed of very smooth, spherical images that can be attributed to water droplets. These classes were originally defined for the 2DS and CIP only because of the difficulty of finding liquid or frozen droplets above 2 mm within the data available to pick from. This class was well trained for all the models. It can be noted that finding water droplets in OAP images can be particularly useful to study mixed-phase clouds. The current method is restricted to sizes greater than a certain pixel threshold, but it should perform better than more established approaches that distinguish water droplets by using the circularity parameter. The obtained PSDs are presented in Figure 18.

The 2DS results for the specific and global models are remarkably similar (Figures 18a and 18b). However, because there were too few big water droplets in the CIP training set, the CIP-specific model failed to detect larger droplets, whereas the global model could. In addition, it could find WD in the PIP data but not in the HVPS. The differences in performance between the specific and global models highlight the importance of having a diverse training dataset. By incorporating data from multiple sources, the global model was able to better generalize and detect larger water droplets across different instruments.

### 3.2.9 Artefact classes : Shattered and Diffracted Particles

Two artifact classes were originally defined for the 2DS (Dif) and the HVPS (SP). Their purpose is to remove artifacts that remain after routine artifact treatments. In the case of diffracted images, these are very specific to the 2DS. The shattered





**Figure 19.** Total particle size distributions for each OAP obtained with the ICE GENESIS data set for the Dif (a and b) and FP (c and d) classes, and identified with specific (a and c) and the global model (b and d). The black points show the distribution points above the photodiode array length.

particles that were found in the HVPS data is significantly different compared to what is usually denoted as shattering in the literature (Korolev and Isaac, 2005). They are not the very small particles that appear in high amounts in the size distribution and that do not affect the current methodology since a relatively high size threshold is used. Instead, this class refers to rather

large particles which appear as a 'cloud of particles' on a single image. The corresponding size distributions are shown in Figure 19.

The Dif PSD obtained with the 2DS specific model is plotted in Figure 19a and exhibits the same shape as the PSD resulting from the use of the global model on the same data (see 2DS curve in Figure 19b). However, the values are much lower in the case of the global model, which shows in the total concentration fraction varying from 7.59 % for the specific model to 4.73 %



for the global model (see Table 1). The application of the global model to the data gathered with other probes yielded very few images in the Dif class, resulting in 0.78, 0.06, and 0.05 % of the total concentration for the CIP, PIP, and HVPS, respectively. For the FP class, the specific model for the HVPS identifies more FP compared to the global model, with 1.79 against 1.28 %. However, the corresponding curves exhibit a similar shape once more (see HVPS PSD in Figures 19c and d). The global model identified a significant amount of FP for the CIP probe that amounts to 3.23 % of the total measured concentration for this instrument. A visual examination of the corresponding pictures demonstrated that these identifications were, in fact, accurate. The singularly high number of FP particles within the CIP compared to the other instruments may be attributed to differences in optics, electronics, and the design of its arms with respect to the position of its laser beam. Compared to the probe with the most similar optics and electronics, the arms of the PIP extend outwards, whereas the arms of the CIP are parallel, which could produce more of these shattering events. In any case, the FP PSD deserves further investigation for the CIP.

## 4 Conclusions

In the present article, morphological data from OAPs was used in a new way. First, three new CNNs were trained, two of which were specific models for the CIP and HVPS, and the last one was a global model that can be used on all OAPs. Two CNNs that were previously developed for the 2DS and the PIP were tested and improved. A methodology was presented to obtain size distributions specific to morphology by combining particle size distributions with CNN models for particle identification.

Then, the ICE GENESIS data set, which comprises all four OAPs, was presented. The morphology-specific PSDs were described and discussed at a high level of detail for each probe and for both specific and global models. For compact particles, columns, combinations of bullets and columns, hexagonal planar crystals, and water droplets, the global model had the advantage of unifying the results obtained from the different instruments compared to the specific models. For capped columns and complex assemblages, the global model was able to extrapolate, with relative success, the classes of the higher resolution probes to the coarser resolution ones. However, major uncertainty about the accuracy of these classes remains when used on PIP or HVPS images, especially for capped columns. For the present data set, these two classes are of minor importance in the HVPS and PIP size range. For the artifact classes, namely diffracted and shattered particles, using the global model decreased the number of images identified for the specific probes for which these classes were originally defined. However, it revealed a high number of shattered particles within the CIP images. This constitutes, in itself, an important result of the study. Further investigations on other data sets will help to determine the limitations and strengths of the global model in classifying different types of particles in OAP images.

The study demonstrated the effectiveness of utilizing CNN models for morphological data interpretation and, in particular, the advantages of using a single global CNN model. The obtained morphology-specific particle size distributions represent a major step in the characterization of atmospheric ice and water particles. A few improvement areas could be identified in the process of examining these spectra. With the training methodology now perfected, these can be rectified through data assimilation, further enhancing the accuracy and reliability of the global CNN model in analyzing OAP data.





The developed models can be applied to decades of OAP data. Since they were developed, these instruments have been the cornerstone of in-situ aircraft measurements of clouds and the gold standard for cloud model outputs. By reducing 50 years of hydrometeor observation across the world to their most pertinent features (size, shape, and concentration), significant im-

provements in the current understanding of cloud processes can likely be achieved. However, putting morphology at the center of ice cloud observation, requires rethinking how ice clouds are conceptualized and modelled. The differentiation of cloud and precipitation particles into several interacting populations brings forward the complexity of the interactions between different hydrometeor types. During collisions between two hydrometeors, breakup, riming, or aggregation may happen depending on their respective types and sizes. These are important events that trigger secondary ice production and influence precipitation

rates. This approach therefore has the potential to improve the understanding of precipitation patterns, and cloud life cycles, for example.

Convolutional neural networks are semi-supervised learning algorithms that deduce image features that can characterize individual morphological classes from manually sorted data. Under the condition that the training is successful, the obtained classification model can be considered as an as objective class definition tool that matches the human ability beyond any worded

description. The presented models are efficient and were the first to be developed for optical array probes. For these reasons, the training sets gathered at the Laboratoire de Météorologie Physique may be considered the first reference data sets after manual inspections and validation by other ice microphysics specialists from different research facilities around the globe. The data set can of course be expanded and evaluated regularly. This will allow for harmonized, consistent, and comparable results across different research teams utilizing automatic classification tools. Finally, sharing the training data will promote collaboration

and exchanges in the fields of ice microphysics and cloud research. With the recent advances in artificial intelligence, the authors believe that collaboration and transparency in data sharing should be promoted, and that, collectively, scientists can create the next generation analysis tools. The present contribution already showed that such developments are possible in the field of cloud microphysics with ice particle images.

*Code and data availability.* Training and testing data (labeled binary images) and Python codes can be made publicly available upon request

to the authors https://github.com/LJaffeux/JAFFEUXetalAMT2024.

*Competing interests.* No competing interests



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
