# Peer review of "Convolutional Neural Networks for specific and merged data sets of Optical Array Probe images: Compatibility of retrieved morphology dependent size distributions."

_EGUsphere, 2024_

## Referee Comment (RC2)

**Review of "Ice crystal images from optical array probes: Compatibility of morphology specific size distributions, retrieved with specific and global Convolutional Neural Networks for HVPS, PIP, CIP, and 2DS" – Jaffeux et al. (2024)**

This study leverages optical array probe data from a recent field campaign to build, test, and apply convolutional neural networks for the identification of ice crystal morphology. Ice crystal morphology in clouds is a particularly useful characteristic, as it can shed light on the evolution of ice crystals. Until recently, identify ice crystal morphology from in situ data has been thwarted by the time and effort needed to manually characterize the data. Recent advances in machine learning have made it possible to build advanced models that can complete the categorization processes in a faction of the time that it takes a human to do so and with very high accuracy.

This work demonstrates this approach using data from several probes to build convolutional neural networks (CNNs), with reasonable accuracy. However, the novelty of the work as well as the presentation of the research is lacking, or at least needs to be better explained. As such, this paper needs to undergo rather significant revisions before it can proceed with the review process.

**Major Comments:**

1) Regarding the CNNs, there is almost no information on what was actually done. CNNs represent a range of possible machine learning algorithms. These details are absolutely needed for the work conducted in this paper to be reproducible, as well as for the reviewers and readers to determine if the approach is viable for the application.
2) The novelty of this work is lacking, and in fact many recent studies that have conducted similar work are not even mentioned. These studies need to be discussed in the introduction (in general, the introduction lacks sufficient information on the problem at hand and how others have tried to resolve it). These studies also need to be discussed in the context of the new findings and thus to show what is new/different/novel in the current work. Recent works of relevance that have applied CNNs to this exact problem include, but are not limited to, Pryzybylo et al. (2022), Zhang et al. (2023), and Schmitt et al. (2024a,b). These papers are mentioned in passing but not highlighted. Mainly, the introduction really needs to highlight what has been done in this field and explain the novelty of the current work.

**Minor Comments:**

1) Acronyms need to be defined the first time they are used.
2) Please consider having a native English-speaking person review the paper for grammar.

**References**

Przybylo, V. M., K. J. Sulia, C. G. Schmitt, and Z. J. Lebo, 2022: Classification of Cloud Particle Imagery from Aircraft Platforms Using Convolutional Neural Networks. *J. Atmos. Oceanic Technol.*, **39**, 405–424, https://doi.org/10.1175/JTECH-D-21-0094.1.

Schmitt, C. G., E. Järvinen, M. Schnaiter, D. Vas, L. Hartl, T. Wong, and M. Stuefer, 2024b: Classification of ice particle shapes using machine learning on forward light scattering images. *Artif. Intell. Earth Syst.*, https://doi.org/10.1175/AIES-D-23-0091.1, in press.

Schmitt, C. G., D. Vas, M. Schnaiter, E. Järvinen, L. Hartl, T. Wong, V. Cassella, and M. Stuefer, 2024a: Microphysical Characterization of Boundary Layer Ice Particles: Results from a 3-Year Measurement Campaign in Interior Alaska. *J. Appl. Meteor. Climatol.*, **63**, 699–716, https://doi.org/10.1175/JAMC-D-23-0190.1.

Zhang, R., Xiao, H., Gao, Y. *et al.* Shape Classification of Cloud Particles Recorded by the 2D-S Imaging Probe Using a Convolutional Neural Network. *J. Meteorol. Res.*, **37**, 521–535 (2023). https://doi.org/10.1007/s13351-023-2146-2.

---

## Author Response (AR1)

**Author's response to public review:**

The title was changed to : Convolutional Neural Networks for specific and merged data sets of Optical Array Probe images : Compatibility of retrieved morphology dependent size distributions.

The motivation of this change is to differentiate the article from other publications which focus on training classification algorithms and focus on the second part of the paper. By doing this we hope to clear some of the confusion concerning the novelty of the article.

The abstract and the introductions have been reworked slightly to insist on the novelty of the work, discuss additionnal references of studies that used similar methodologies, and show the usefulness of our work for the scientific community.

Subsection 2.1 has been renamed "Morphological classes" (earlier "Data and Morphological classes").

Additional *CNN* information has been added in section 2 in the form of a schematic as was stated in reply to RC1 (https://doi.org/10.5194/egusphere-2024-1910-AC1). This part of the article is easily reproducible thanks to the open access Github repository that contains all used images used for the CNN models.

Acronyms are now defined from the beginning (which includes the abstract). In particular, probe and institute names will be fully written, the first time they are used.

Grammar has been checked once more.

---

## Author Response (AR2)

**Response to reviewer #2:**

The introduction has been extended, in particular in the fifth paragraph, with some context about machine learning in atmospheric sciences. It now explains more clearly the advantages of ML to perform ice particle classification based on OAP images. In addition, the abstract has been improved to more clearly reflect what is the content of the article.

---

## Author Response (AR3)

Dear Wiebke Frey,

Thank you for giving us the opportunity to improve the introduction.
We have taken your suggestions into consideration and in some cases they revealed the need to clarify the text, go into more detail about the CNN technique to demonstrate why it is a significant improvement over older methodologies, and more directly state the motivations of this study. The quoted lines refer to the numbering of the new track change version.

**Response to Editor's Public Justification:**

*(line numbers refer to the track change version)*
*line 1: "...and allows a more refined processing..."*
*This study allows processing? I think, 'allows' is not the right word here, please rephrase.*

The full sentence was changed to :

"This study addresses the challenges of ice particle morphology classification from images of Optical Array Probes (OAP) and proposes a more refined processing to enable better interpretation of observational data."

**For the following comments substantial changes were made.**

line 64 : "CNNs ... automatically identify patterns and features ..." How do CNNs differ then from feature based algorithms (line 60)? Please clarify this.

The series of paragraphs starting at line 55 and continuing up to line 89 have been rearranged. Despite it being done more specifically in Section 2 for the models trained in the study, the general principles of the CNN methodology are shortly presented in order to distinguish them from feature-based approaches. In addition, it demonstrates more clearly why CNNs are more effective.

lines 82-85: I think this sentence is not entirely clear, could you please rephrase or expand the explanation to make it better understandable.

The paragraphs starting at line 90 and finishing at line 108 have been reworked to more clearly express the motivations of the study, which was the intent of this sentence. The sentence itself has been made into two sentences, completed with further explanations, and the remainder of the paragraph has been redirected towards this one message.

**The following suggestions were applied to the manuscript :**

line 13: "from whatever OAP dataset" - remove 'whatever'

line 16/17: please remove "and thus extends..." I think that goes a bit too far.

line 74: remove "also"

line 47-49: Again, why do you change this sentence and its meaning? In the new version, it is not clear how knowledge of morphology is translating to improved weather forecast (do weather forecast model

treat ice crystal habits at all?). Please change back to old version.

line 71/71: "... sample volumes, however..." Please make it two sentences (...sample volume. However,...)

line 79: "This current study " -> The current study

line 90/91: remove sentence, "grey" literature; same for line 312 - grey literature should be avoided. To the least, the conference name and place should be given in the references.

**This change was reverted back keeping "two order of magnitudes in size".**

line 94: why the change from "most of" the size range to "the full"?